# EFFICIENT EXPLORATION VIA STATE MARGINAL MATCHING

## ABSTRACT

Reinforcement learning agents need to explore their unknown environments to solve the tasks given to them. The Bayes optimal solution to exploration is intractable for complex environments, and while several exploration methods have been proposed as approximations, it remains unclear what underlying objective is being optimized by existing exploration methods, or how they can be altered to incorporate prior knowledge about the task. Moreover, it is unclear how to acquire a single exploration strategy that will be useful for solving multiple downstream tasks. We address these shortcomings by learning a single exploration policy that can quickly solve a suite of downstream tasks in a multi-task setting, amortizing the cost of *learning to explore*. We recast exploration as a problem of *State Marginal Matching* (SMM), where we aim to learn a policy for which the state marginal distribution matches a given target state distribution, which can incorporate prior knowledge about the task. We optimize the objective by reducing it to a two-player, zero-sum game between a state density model and a parametric policy. Our theoretical analysis of this approach suggests that prior exploration methods do not learn a policy that does distribution matching, but acquire a replay buffer that performs distribution matching, an observation that potentially explains prior methods' success in single-task settings. On both simulated and real-world tasks, we demonstrate that our algorithm explores faster and adapts more quickly than prior methods.[1]

## 1 INTRODUCTION

Reinforcement learning (RL) algorithms must be equipped with exploration mechanisms to effectively solve tasks with limited reward signals. These tasks arise in many real-world applications where providing human supervision is expensive. The inability of current RL algorithms to adequately explore limits their applicability to long-horizon control tasks.

A wealth of prior work has studied exploration for RL. While, in theory, the Bayes-optimal exploration strategy is optimal, it is intractable to compute exactly, motivating work on tractable heuristics for exploration. Exploration methods based on random actions have limited ability to cover a wide range of states. More sophisticated techniques, such as intrinsic motivation, accelerate learning in the single-task setting. However, these methods have two limitations. First, they do not explicitly define an objective to quantify "good exploration," but rather argue that exploration arises implicitly through some iterative procedure. Lacking a well-defined optimization objective, it remains challenging to understand what these methods are doing and why they work. Similarly, the lack of a metric to quantify exploration, even if only for evaluation, makes it challenging to compare exploration methods and assess progress in this area. The second limitation is that these methods target the single-task setting. Because these methods aim to converge to the optimal policy for a particular task, it is challenging to repurpose these methods to solve multiple tasks.

We address these shortcomings by considering a multi-task setting, where many different reward functions can be provided for the same set of states and dynamics. Rather than exploring from scratch for each task, we aim to learn a *single, task-agnostic exploration policy* that can be adapted to many possible downstream reward functions, amortizing the cost of learning to explore. This exploration

---

[1]Videos and code: https://sites.google.com/view/smm-anonymous

policy can be viewed as a prior on the policy for solving downstream tasks. Learning will consist of two phases: during training, we acquire this task-agnostic exploration policy; during testing, we use this exploration policy to quickly explore and maximize the task reward.

Learning a single exploration policy is considerably more difficult than doing exploration throughout the course of learning a single task. The latter is done by intrinsic motivation (Pathak et al., 2017; Tang et al., 2017; Oudeyer et al., 2007) and count-based exploration methods (Bellemare et al., 2016), which can effectively explore to find states with high reward, at which point the agent can decrease exploration and increase exploitation of those high-reward states. While these methods perform efficient exploration for learning a single task, the policy at any particular iteration is not a good exploration policy. For example, the final policy at convergence would only visit the high-reward states discovered for the current task.

*What objective should be optimized to obtain a good exploration policy?* We recast exploration as a problem of State Marginal Matching: given a desired state distribution, we learn a mixture of policies for which the state marginal distribution matches this desired distribution. Without any prior information, this objective reduces to maximizing the marginal state entropy $\mathcal{H}[s]$, which encourages the policy to visit as many states as possible. The distribution matching objective also provides a convenient mechanism to incorporate prior knowledge about the task, whether in the form of safety constraints that the agent should obey; preferences for some states over other states; reward shaping; or the relative importance of each state dimension for a particular task.

We also propose an algorithm to optimize the State Marginal Matching (SMM) objective. First, we reduce the problem of SMM to a two-player, zero-sum game between a policy player and a density player. We find a Nash Equilibrium for this game using *fictitious play* (Brown, 1951), a classic procedure from game theory. Our resulting algorithm iteratively fits a state density model and then updates the policy to visit states with low density under this model. Our analysis of this approach sheds light on prior work on exploration. In particular, while the *policy* learned by existing exploration algorithms does not perform distribution matching, the *replay buffer* does, an observation that potentially explains the success of prior methods. On both simulated and real-world tasks, we demonstrate that our algorithm explores more effectively and adapts more quickly to new tasks than state-of-the-art baselines.

## 2 RELATED WORK

Most prior work on exploration has looked at exploration bonuses and intrinsic motivation. One class of exploration methods uses prediction error of some auxiliary task as an exploration bonus, which provides high (intrinsic) reward in states where the predictive model performs poorly (Pathak et al., 2017; Oudeyer et al., 2007; Schmidhuber, 1991; Houthooft et al., 2016; Burda et al., 2018). Another set of approaches (Tang et al., 2017; Bellemare et al., 2016; Schmidhuber, 2010) directly encourage the agent to visit novel states. While all methods effectively explore during the course of solving a single task (Taïga et al., 2019), the policy obtained at convergence is often not a good exploration policy (see Section 4). In contrast, our method converges to a highly-exploratory policy by maximizing state entropy in the training objective (Eq. 2).

Many exploration algorithms can be classified by whether they explore in the space of actions, policy parameters, goals, or states. Common exploration strategies including $\epsilon$-greedy and Ornstein–Uhlenbeck noise (Lillicrap et al., 2015), as well as standard MaxEnt RL algorithms (Ziebart, 2010; Haarnoja et al., 2018), explore in the action space. Recent work (Fortunato et al., 2017; Plappert et al., 2017) shows that adding noise to the parameters of the policy can result in good exploration. Most closely related to our work are methods that perform exploration in the space of states or goals (Colas et al., 2018; Held et al., 2017; Nair et al., 2018; Pong et al., 2019; Hazan et al., 2018). In fact, Hazan et al. (2018) consider the same State Marginal Matching objective that we examine and propose a similar algorithm. In relation to Hazan et al. (2018), our main contributions are (1) empirically showing that exploration based on state-entropy is competitive with existing state-of-the-art exploration methods, and (2) explaining how existing exploration methods based on prediction error are *implicitly* maximizing this state-entropy objective. In Appendix C.1, we also discuss how goal-conditioned RL (Kaelbling, 1993; Schaul et al., 2015) can be viewed as a special case of State Marginal Matching when the goal-sampling distribution is learned jointly with the policy.

The problems of exploration and meta-reinforcement learning are tightly coupled. Meta-reinforcement learning algorithms (Duan et al., 2016; Finn et al., 2017; Rakelly et al., 2019; Mishra et al., 2017) must perform effective exploration if they hope to solve a downstream task. Some prior work has explicitly looked at the problem of learning to explore (Gupta et al., 2018; Xu et al., 2018). Our problem statement is similar to meta-learning, in that we also aim to learn a policy as a prior for solving downstream tasks. However, whereas meta-RL requires a distribution of task reward functions, our method will require only a single target state marginal distribution. Due to the simpler problem assumptions and training procedure, our method may be easier to apply in real-world domains.

Related to our approach are standard maximum *action* entropy algorithms (Haarnoja et al., 2018; Kappen et al., 2012; Rawlik et al., 2013; Ziebart et al., 2008; Theodorou & Todorov, 2012). While these algorithms are referred to as *MaxEnt RL*, they are maximizing entropy over actions, not states. These algorithms can be viewed as performing inference on a graphical model where the likelihood of a trajectory is given by its exponentiated reward (Toussaint & Storkey, 2006; Levine, 2018; Abdolmaleki et al., 2018). While distributions over trajectories induce distributions over states, computing the exact relationship requires integrating over all possible trajectories, an intractable problem for most MDPs. A related but distinct class of *relative* entropy methods use a similar entropy-based objective to limit the size of policy updates (Peters et al., 2010; Schulman et al., 2015).

Finally, the idea of distribution matching has been employed successfully in imitation learning (Ziebart et al., 2008; Ho & Ermon, 2016; Finn et al., 2016; Fu et al., 2017). Similar to some inverse RL algorithms (Ho & Ermon, 2016; Fu et al., 2018), our method iterates between learning a policy and learning a reward function, though our reward function is obtained via a density model instead of a discriminator. While inverse RL algorithms assume access to expert trajectories, we instead assume access to the density of the target state marginal distribution. In many realistic settings, such as robotic control with many degrees of freedom, providing fully-specified trajectories may be much more challenging than defining a target state marginal distribution. The latter only requires some aggregate statistics about expert behavior, and does not even need to be realizable by any policy.

In summary, our work unifies prior exploration methods as performing approximate distribution matching, and explains how state distribution matching can be performed properly. This perspective provides a clearer picture of exploration, and this observation is useful particularly because many of the underlying ingredients, such as adversarial games and density estimation, have seen recent progress and therefore might be adopted to improve exploration methods.

## 3 STATE MARGINAL MATCHING

In this section, we propose the State Marginal Matching problem as a principled objective for *learning to explore*, and offer an algorithm for optimizing it. We consider a parametric policy $\pi_\theta \in \Pi \triangleq \{\pi_\theta \mid \theta \in \Theta\}$ that chooses actions $a \in \mathcal{A}$ in a Markov Decision Process (MDP) $\mathcal{M}$ with fixed episode lengths $T$, dynamics distribution $p(s_{t+1} \mid s_t, a_t)$, and initial state distribution $p_0(s)$. The MDP $\mathcal{M}$ together with the policy $\pi_\theta$ form an implicit generative model over states. We define the *state marginal distribution* $\rho_\pi(s)$ as the probability that the policy visits state $s$:

$$\rho_\pi(s) \triangleq \mathbb{E}_{\substack{s_1 \sim p_0(S), \\ a_t \sim \pi_\theta(A|s_t) \\ s_{t+1} \sim p(S|s_t,a_t)}} \left[ \frac{1}{T} \sum_{t=1}^{T} \mathbb{1}(s_t = s) \right]$$

We emphasize that $\rho_\pi(s)$ is not a distribution over trajectories, and is not the stationary distribution of the policy after infinitely many steps, but rather the distribution over states visited in a finite-length episode.[2] We also note that any trajectory distribution matching problem can be reduced to a state marginal matching problem by augmenting the current state to include all previous states.

We assume that we are given a target distribution $p^*(s)$ over states $s \in \mathcal{S}$ that encodes our belief about the tasks we may be given at test-time. For example, a roboticist might assign small values of $p^*(s)$ to states that are dangerous, regardless of the desired task. Alternatively, we might also learn $p^*(s)$ from data about human preferences (Christiano et al., 2017). For goal-reaching tasks, we can analytically derive the optimal target distribution (Appendix C). Given $p^*(s)$, our goal is to find a

---

[2] $\rho_\pi(s)$ approaches the policy's stationary distribution in the limit as the episodic horizon $T \to \infty$.

Figure 1: **State Marginal Matching**: *(Left)* Our goal is to learn a policy whose distribution over states (blue histogram) matches some target density (black line). Our algorithm iteratively increases the reward on states visited too infrequently (green arrow) and decreases the reward on states visited too frequently (red arrow). *(Center)* At convergence, these two distributions are equal. *(Right)* For complex target distributions, we use a mixture of policies $\rho_\pi(s) = \int \rho_{\pi_z}(s)p(z)dz$. (See Appendix B.)

parametric policy that is "closest" to this target distribution, where we measure discrepancy using the Kullback-Leibler (KL) divergence:

$$\max_{\pi \in \Pi} \mathcal{F}(\rho_\pi(s), p^*(s)) \triangleq \max_{\pi \in \Pi} -D_{\mathrm{KL}}(\rho_\pi(s) \parallel p^*(s))$$

$$= \max_{\pi \in \Pi} \mathbb{E}_{s \sim \rho_\pi(s)}[\log p^*(s) - \log \rho_\pi(s)] \tag{1}$$

$$= \max_{\pi \in \Pi} \mathbb{E}_{s \sim \rho_\pi(s)}[\log p^*(s)] + \mathcal{H}_\pi[s] \tag{2}$$

This is the same objective as in Hazan et al. (2018). Note that we use the reverse-KL (Bishop, 2006), which is mode-seeking (i.e., exploratory). We show in Appendix C that the policies obtained via State Marginal Matching provide an optimal exploration strategy for a particular distribution over reward functions. To gain intuition for the State Marginal Matching objective, we decomposed it in two ways. In Equation 2, we see that State Marginal Matching is equivalent to maximizing the reward function $r(s) \triangleq \log p^*(s)$ while simultaneously maximizing the entropy of states. Note that, unlike traditional MaxEnt RL algorithms (Ziebart et al., 2008; Haarnoja et al., 2018), we regularize the entropy of the state distribution, not the conditional distribution of actions given states, which results in exploration in the space of states rather than in actions. Moreover, Equation 1 suggests that State Marginal Matching maximizes a pseudo-reward $r(s) \triangleq \log p^*(s) - \log \rho_\pi(s)$, which assigns positive utility to states that the agent visits too infrequently and negative utility to states visited too frequently (see Figure 1). We emphasize that *maximizing this pseudo-reward is not a RL problem because the pseudo-reward depends on the policy.*

### 3.1 OPTIMIZING THE STATE MARGINAL MATCHING OBJECTIVE

Optimizing Equation 1 to obtain a single exploration policy is more challenging than standard RL because the reward function itself depends on the policy. To break this cyclic dependency, we introduce a parametric state density model $q_\psi(s) \in Q \triangleq \{q_\psi \mid \psi \in \Psi\}$ to approximate the policy's state marginal distribution, $\rho_\pi(s)$. We assume that the class of density models $Q$ is sufficiently expressive to represent every policy:

**Assumption 1.** *For every policy $\pi \in \Pi$, there exists $q \in Q$ such that $D_{KL}(\rho_\pi(s) \parallel q(s)) = 0$.*

Under this assumption, optimizing the policy w.r.t. this approximate distribution $q(s)$ will yield the same solution as Equation 1 (see Appendix A for the proof):

**Proposition 3.1.** *Let policies $\Pi$ and density models $Q$ satisfying Assumption 1 be given. For any target distribution $p^*$, the following optimization problems are equivalent:*

$$\max_\pi \mathbb{E}_{\rho_\pi(s)}[\log p^*(s) - \log \rho_\pi(s)] = \max_\pi \min_q \mathbb{E}_{\rho_\pi(s)}[\log p^*(s) - \log q(s)] \tag{3}$$

Solving the new max-min optimization problem is equivalent to finding the Nash equilibrium of a two-player, zero-sum game: a *policy player* chooses the policy $\pi$ while the *density player* chooses the density model $q$. To avoid confusion, we use *actions* to refer to controls $a \in \mathcal{A}$ output by the policy $\pi$ in the traditional RL problem and *strategies* to refer to the decisions $\pi \in \Pi$ of the policy player and decisions $q \in Q$ of the density player. The Nash existence theorem (Nash, 1951) proves that such a stationary point always exists for such a two-player, zero-sum game.

One common approach to saddle point games is to alternate between updating player A w.r.t. player B, and updating player B w.r.t. player A. However, games such as Rock-Paper-Scissors illustrate that such a greedy approach is not guaranteed to converge to a stationary point. A slight variant,

---

**Algorithm 1** Learning to Explore via Fictitious Play

---

**Input:** Target distribution $p^*(s)$
Initialize policy $\pi(a \mid s)$, density model $q(s)$, and replay buffer $\mathcal{B}$.
**while** not converged **do**
$\quad q^{(m)} \leftarrow \arg\max_q \mathbb{E}_{s \sim \mathcal{B}^{(m-1)}} \left[\log q(s)\right]$
$\quad \pi^{(m)} \leftarrow \arg\max_\pi \mathbb{E}_{s \sim \rho_\pi(s)} \left[r(s)\right]$ where $r(s) \triangleq \log p^*(s) - \log q^{(m)}(s)$
$\quad \mathcal{B}^{(m)} \leftarrow \mathcal{B}^{(m-1)} \cup \{(s_t, a_t, s_{t+1})\}_{t=1}^T$ with new transitions $\{(s_t, a_t, s_{t+1})\}_{t=1}^T$ sampled from $\pi^{(m)}$
**return** historical policies $\{\pi^{(1)}, \cdots, \pi^{(m)}\}$

---

Algorithm 1: An algorithm for optimizing the State Marginal Matching objective (Equation 1). The algorithm iterates between (1) fitting a density model $q^{(m)}$ and (2) training the policy $\pi^{(m)}$ with a RL objective to optimize the expected return w.r.t. the updated reward function $r(s)$. The algorithm returns the collection of policies from each iteration, which do distribution matching in aggregate.

*fictitious play* (Brown, 1951) does converge to a Nash equilibrium in finite time (Robinson, 1951; Daskalakis & Pan, 2014). At each iteration, each player chooses their best strategy in response to the *historical average* of the opponent's strategies. In our setting, fictitious play alternates between fitting the density model to the historical average of policies (Equation 4), and updating the policy with RL to minimize the log-density of the state, using a historical average of the density models (Equation 5):

$$q_{m+1} \leftarrow \arg\max_q \mathbb{E}_{s \sim \bar{\rho}_m(s)} \left[\log q(s)\right] \qquad \text{where} \quad \bar{\rho}_\pi^{(m)}(s) \triangleq \frac{1}{m} \sum_{i=1}^m \rho_{\pi_i}(s) \quad (4)$$

$$\pi_{m+1} \leftarrow \arg\max_\pi \mathbb{E}_{s \sim \rho_\pi(s)} \left[\log p^*(s) - \log \bar{q}_m(s)\right] \qquad \text{where} \quad \bar{q}_m(s) \triangleq \frac{1}{m} \sum_{i=1}^m q_i(s) \quad (5)$$

Crucially, the exploration policy is not the last policy, $\pi_{m+1}$, but rather the historical average policy:

**Definition 3.1.** A *historical average policy* $\bar{\pi}(a \mid s)$, parametrized by a collection of policies $\pi_1, \cdots, \pi_m$, is a policy that randomly samples one of the policy iterates $\pi_i \sim \text{Unif}[\pi_1, \cdots, \pi_m]$ at the start of each episode and takes actions according to that policy for each step in the episode. A new policy is sampled for the next episode.

We summarize the resulting algorithm in Algorithm 1. In practice, we can efficiently implement Equation 4 and avoid storing the policy parameters from every iteration by instead storing sampled states from each iteration.[3] We cannot perform the same trick for Equation 5, and instead resort to approximating the historical average of density models with the most recent iterate. Algorithm 1 looks similar to prior exploration methods based on prediction-error, suggesting that we might use SMM to understand how these prior methods work.

## 4 WHY DOES PREDICTION-ERROR EXPLORATION WORK?

Exploration methods based on prediction error (Burda et al., 2018; Stadie et al., 2015; Pathak et al., 2017; Schmidhuber, 1991; Chentanez et al., 2005) do not converge to an exploratory policy, even in the absence of extrinsic reward. For example, consider the asymptotic behavior of ICM (Pathak et al., 2017) in a deterministic MDP, such as the Atari games where it was evaluated. At convergence, the predictive model will have zero error in all states, so the exploration bonus is zero – the ICM objective has no effect on the policy at convergence. Similarly, consider the exploration bonus in Pseudocounts (Bellemare et al., 2016): $1/\hat{n}(s)$, where $\hat{n}(s)$ is the (estimated) number of times that state $s$ has been visited. In the infinite limit, each state has been visited infinitely many times, so the Pseudocount exploration bonus also goes to zero — Pseudocounts has no effect at convergence. Similar reasoning can be applied to other methods based on prediction error (Burda et al., 2018; Stadie et al., 2015). More broadly, we can extend this analysis to stochastic MDPs, where we consider an abstract exploration algorithm that alternates between computing some intrinsic reward and performing RL (to convergence) on that intrinsic reward. Existing prediction-error exploration

---

[3]One approach is to maintain an infinite-sized replay buffer, and fit the density to the replay buffer at each iteration. Alternatively, we can replace older samples in a fixed-size replay buffer less frequently such that sampling from $\mathcal{B}$ is uniform over iterations.

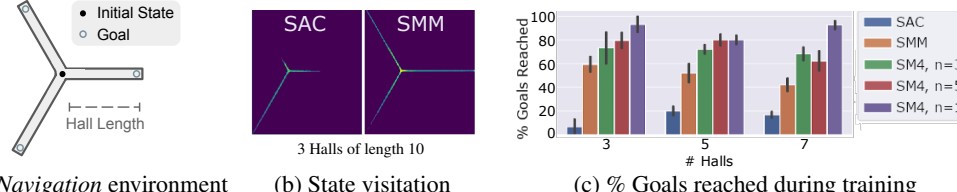

(a) *Navigation* environment     (b) State visitation     (c) % Goals reached during training

Figure 2: **Exploration in State Space (SMM) vs. Action Space (SAC) for *Navigation*: (a)**: A point-mass agent is spawned at the center of $m$ long hallways that extend radially outward, and the target state distribution places uniform probability mass $\frac{1}{m}$ at the end of each hallway. We can vary the length of the hallway and the number of hallways to control the task difficulty. **(b)** A heatmap showing states visited by SAC and SMM during training illustrates that SMM explores a wider range of states. **(c)** SMM reaches more goals than the MaxEnt baseline. SM4 is an extension of SMM that incorporates mixture modelling with $n > 1$ skills (see Appendix B), and further improves exploration of SMM.

methods are all special cases. At each iteration, the RL step solves a fully-observed MDP, which always admits a deterministic policy as a solution (Puterman, 2014). Thus, any exploration algorithm in this class cannot converge to a single, exploratory policy.

Despite these observations, prior methods do excel at solving hard exploration tasks. We draw an analogy to fictitious play to explain their success. While these methods never acquire an exploratory policy, over the course of training they will eventually visit all states. In other words, the *historical average* over policies will visit a wide range of states. Since the replay buffer exactly corresponds to this historical average over states, these methods will obtain a replay buffer with a diverse range of experience, possibly explaining why they succeed at solving hard exploration tasks. Moreover, this analysis suggests a surprisingly simple method for obtaining an exploration from these prior methods: use a mixture of the policy iterates throughout training. The following section will not only compare SMM against prior exploration methods, but also show that this historical averaging trick can be used to improve existing exploration methods.

## 5 SIMULATED EXPERIMENTS

We used simulated control tasks to determine if SMM learns an exploratory policy, to compare SMM to prior exploration methods, and to study the effect of historical averaging. More details can be found in Appendix D, and code will be released upon publication.

**Baselines and Implementation Details**: We compare to a state-of-the-art off-policy MaxEnt RL algorithm, Soft Actor-Critic (SAC) (Haarnoja et al., 2018); an inverse RL algorithm, Generative Adversarial Imitation Learning (GAIL) (Ho & Ermon, 2016); and three exploration methods:

- Count-based Exploration (C), which discretizes states and uses $-\log \hat{\pi}(s)$ as an exploration bonus.
- Pseudo-counts (PC) (Bellemare et al., 2016), which uses the recoding probability as a bonus.
- Intrinsic Curiosity Module (ICM) (Pathak et al., 2017), which uses prediction error as a bonus.

We used SAC as the base RL algorithm for all exploration methods (SMM, C, PC, ICM). To implement SMM, we define the target distribution in terms of the extrinsic environment reward: $p^*(s) \propto \exp(r_{\text{env}}(s))$. We use a variational autoencoder (VAE) to model the density $q(s)$ for both SMM and Pseudocounts (PC). For the GAIL baseline, we generated synthetic expert data by sampling expert states from the target distribution $p^*(s)$ (see Appendix D.2 for details). Results for all experiments are averaged over 4-5 random seeds.

We start with a sanity check: Is exploration in state space (as done by SMM) better than exploration in action space (as done by MaxEnt RL, e.g., SAC)? To study this question, we implemented a 2D *Navigation* environment, shown in Figure 2a. To evaluate each method, we counted the number of hallways that the agent fully explored (i.e., reached the end) during training. Figure 2b shows the state visitations for the three hallway environment, illustrating that SAC only explores one hallway, whereas SMM explores all three. Figure 2c also shows that SMM consistently explores 60% of hallways, whereas SAC rarely visits more than 20% of hallways.

The remaining simulated experiments used the *Manipulation* environment (Plappert et al., 2018), shown in Figure 3a. Our first experiment evaluates whether the exploration policy acquired by SMM allows us to solve downstream tasks more quickly. We defined the target distribution to be uniform over the entire state space (joint + block configuration), with the constraints that we put

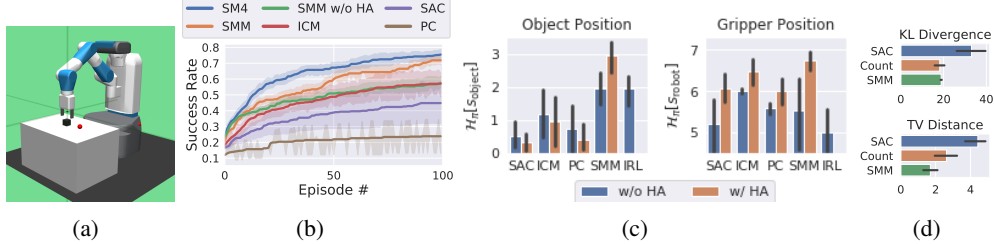

(a)  (b)  (c)  (d)

Figure 3: **Exploration for *Manipulation*.** **(a) Task**: The robot agent controls a single gripper arm to move a block object to a goal location on the table surface. The goal is not observed by the robot, thus requiring the robot to explore by moving the block to different locations on the table. **(b) Test-Time Exploration**: At test-time, we sample goal locations uniformly on the table, and plot the percentage of goals found within $N$ episodes. SMM and its mixture-model variant SM4 (Algorithm 2) both explore faster than the baselines, allowing it to successfully find the goal in fewer episodes. **(c) State Entropy**: After training, we rollout the policy for 1e3 epochs, and record the entropy of the object and gripper positions. SMM achieves higher state entropy than the other methods. Historical averaging also improves the exploration of prior methods. **(d) Non-Uniform Exploration**: We measure the discrepancy between the state marginal distribution, $\rho_\pi(s)$, and a non-uniform target distribution. SMM matches the target distribution better than SAC and is on par with Count. Error bars show std. dev. across 4 random seeds.

low probability mass on states where the block has fallen off the table; that actions should be small; and that the arm should be close to the object. As shown in Figure 3b, SMM adapts substantially more quickly than other exploration methods, achieving a success rate 20% higher than the next best method, and reaching the same level of performance of the next baseline (ICM) in 4x fewer episodes. SMM without historical averaging attains similar performance as the next best baseline (ICM), suggesting that historical averaging is the key ingredient, while the particular choice of prediction error or VAE is less important. We provide further ablation studies of SMM in Appendix B.2.

While historical averaging is necessary to guarantee convergence (§ 3.1), most prior exploration methods do not employ historical averaging, raising the question of whether it is necessary in practice. To answer this question, we compare SMM to three exploration methods. In Figure 3c, we compare the policy obtained at convergence with the historical average of policy iterates over training for each method. We measure how well each method explores by computing the marginal state entropy, which we compute by discretizing the state space.[4] The results show that SMM maximizes state entropy at least as effectively as prior methods, if not better. While this comparison is somewhat unfair, as we measure exploration using the objective that SMM maximizes, none of the methods we compare against propose metrics for exploration that we could use instead. Furthermore, we see that historical averaging is not only beneficial to SMM, but also improves the exploration of prior methods.

In our final simulated experiment, we check whether prior knowledge injected via the target distribution is reflected in the policy obtained from State Marginal Matching. Using the same *Manipulation* environment as above, we modified the target distribution to assign larger probability to states where the block was on the left half of the table than on the right half. In Figure 3d, we measure whether SMM is able to achieve the target distribution by measuring the discrepancy between the block's horizontal coordinate and the target distribution. Compared to the SAC baseline, SMM and the Count baseline are half the distance to the target distribution. No method achieves zero discrepancy, suggesting that future methods could be better at matching state marginals.

## 6 REAL-WORLD EXPERIMENTS

While almost all research on exploration focus on simulated domains, attributes of the real world such as partial observability, nonstationarity, and stochasticity may make the exploration more challenging. The aim of this section is to see if SMM explores effectively on a real-world robotic control task. We used the *D'Claw* (Ahn et al., 2019) robotic manipulator, which is a 3-fingered hand positioned vertically above a handle that it can turn. For all experiments on the *D'Claw*, we used a target distribution that places uniform mass over all object angles $[-180°, 180°]$.

---

[4]Discretization is used only for evaluation, no policy has access to it (except for Count).

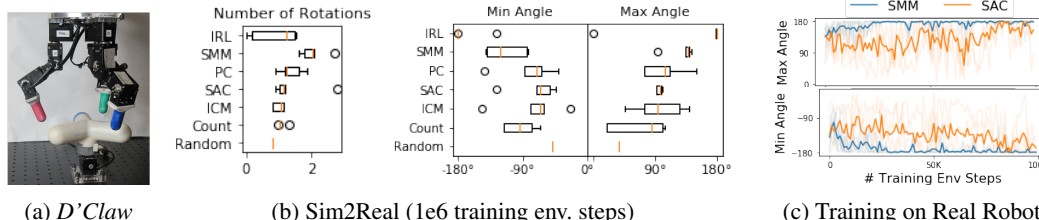

(a) *D'Claw*    (b) Sim2Real (1e6 training env. steps)    (c) Training on Real Robot

Figure 4: **Real-World Exploration**: **(a)** *D'Claw* is a 9-DoF robotic hand (Ahn et al., 2019) that is trained to turn a valve object. **(b) Sim2Real**: We trained each algorithm in simulation, and then measured how far the trained policy rotated the knob on the hardware robot. We also measured the maximum angle that the agent turned the knob in the clockwise and counter-clockwise directions within one episode. **(c) Training on Hardware**: We trained SAC and SMM on the real robot for 1e5 environment steps (about 9 hours in real time), and measured the maximum angle turned throughout training. We see that SMM moves the knob more and visits a wider range of states than SAC. All results are averaged over 4-5 seeds.

In a first experiment, we trained SMM and other baselines in simulation, and then evaluated the acquired exploration policy on the real robot using two metrics: the total number of rotations (in either direction), and the maximum radians turned (in both directions). For each method, we computed the average metric across 100 evaluation episodes. We repeated this process for 5 independent training runs. Figure 4b shows that SMM turns the knob more than the baselines, and it turns the knob to a wider range of angles. To test for statistical significance, we used a 1-sided Student's t-test to test the hypothesis that SMM turned the knob more and to a wider range of angles than SAC. The p-values were all less than 0.05: $p = 0.046$ for number of rotations, $p = 0.019$ for maximum clockwise angle, and $p = 0.001$ for maximum counter-clockwise angle.

In our second experiment, we investigated whether it was possible to learn an exploration policy directly in the real world, without the need for a simulator. Learning to explore in the real world is quite important, as building faithful simulators of complex systems is challenging. The physical constraints of the real robot make data efficiency paramount, suggesting that learning to explore will require an effective exploration strategy. In Figure 4c, we plot the range of angles that the policy explores *throughout* training. Not only does SMM explore a wider range of angles than SAC, but its ability to explore increases throughout training, suggesting that the SMM objective is correlated with real-world metrics of exploration.

In summary, the results in this section suggests that exploration techniques may actually be useful in the real world, which may encourage future work to study exploration methods on real-world tasks.

## 7 DISCUSSION

In this paper, we introduced a formal objective for exploration. While it is often unclear what existing exploration algorithms will converge to, our State Marginal Matching objective has a clear solution: at convergence, the policy should visit states in proportion to their density under a target distribution. Not only does this objective encourage exploration, it also provides human users with a flexible mechanism to bias exploration towards states they prefer and away from dangerous states. Upon convergence, the resulting policy can thereafter be used as a prior in a multi-task setting, amortizing exploration and enabling faster adaptation to new, potentially sparse, reward functions. The algorithm we proposed looks quite similar to previous exploration methods based on prediction error, suggesting that those methods are also performing some form of distribution matching. However, by deriving our method from first principles, we note that these prior methods omit a crucial historical averaging step, without which the algorithm is not guaranteed to converge. Experiments on both simulated and real-world tasks demonstrated how SMM learns to explore, enabling an agent to efficiently explore in new tasks provided at test time.

In future work, we aim to study connections between inverse RL, MaxEnt RL and state marginal matching, all of which perform some form of distribution matching. Empirically, we aim to scale to more complex tasks by parallelizing the training of all mixture components simultaneously. Broadly, we expect the state distribution matching problem formulation to enable the development of more effective and principled RL methods that reason about distributions rather than individual states.

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

## A  PROOFS

*Proof of Proposition 3.1.* Note that

$$\mathbb{E}_{\rho_\pi(s)}[\log p^*(s) - \log q(s)] = \mathbb{E}_{\rho_\pi(s)}[\log p^*(s) - \log \rho_\pi(s)] + D_{\text{KL}}(\rho_\pi(s) \parallel q(s)). \quad (6)$$

By Assumption 1, $D_{\text{KL}}(\rho_\pi(s) \parallel q(s)) = 0$ for some $q \in Q$, so we obtain the desired result:

$$\max_\pi \left( \min_q \mathbb{E}_{\rho_\pi(s)}[\log p^*(s) - \log q(s)] \right) = \max_\pi \left( \mathbb{E}_{\rho_\pi(s)}[\log p^*(s) - \log \rho_\pi(s)] + \min_q D_{\text{KL}}(\rho_\pi(s) \parallel q(s)) \right)$$

$$= \max_\pi \mathbb{E}_{\rho_\pi(s)}[\log p^*(s) - \log \rho_\pi(s)]. \quad \square$$

## B  BETTER MARGINAL MATCHING WITH MIXTURES OF MIXTURES

This section introduces an extension of SMM, SM4, that incorporates mixure modelling. Given the challenging problem of exploration in large state spaces, it is natural to wonder whether we can accelerate exploration by automatically decomposing the potentially-multimodal target distribution into a mixture of "easier-to-learn" distributions and learn a corresponding set of policies to do distribution matching for each component. Note that the mixture model we introduce here is orthogonal to the historical averaging step discussed before. Using $\rho_{\pi_z}(s)$ to denote the state distribution of the policy conditioned on the latent variable $z \in \mathcal{Z}$, the state marginal distribution of the mixture of policies is

$$\rho_\pi(s) = \int_\mathcal{Z} \rho_{\pi_z}(s)p(z)dz = \mathbb{E}_{z\sim p(z)}\left[\rho_{\pi_z}(s)\right], \quad (7)$$

where $p(z)$ is a latent prior. As before, we will minimize the KL divergence between this mixture distribution and the target distribution. Using Bayes' rule to re-write $\rho_\pi(s)$ in terms of conditional probabilities, we obtain the following optimization problem:

$$\max_{(\pi_z)_{z\in\mathcal{Z}}} \mathbb{E}_{\substack{z\sim p(z)\\s\sim\rho_{\pi_z}(s)}} \left[ \log \frac{p^*(s)}{\frac{\rho_{\pi_z}(s)p(z)}{p(z|s)}} \right] = \mathbb{E}_{\substack{z\sim p(z)\\s\sim\rho_{\pi_z}(s)}} \left[ \underbrace{\log p^*(s)}_{(a)} - \underbrace{\log \rho_{\pi_z}(s)}_{(b)} + \underbrace{\log p(z \mid s)}_{(c)} - \underbrace{\log p(z)}_{(d)} \right]$$
$$(8)$$

Intuitively, this says that the agent should go to states (a) with high density under the target state distribution, (b) where this agent has not been before, and (c) where this agent is clearly distinguishable from the other agents. The last term (d) says to explore in the space of mixture components $z$. This decomposition bears a resemblance to the mutual-information objectives in recent work (Achiam et al., 2018; Eysenbach et al., 2018; Co-Reyes et al., 2018). Thus, one interpretation of our work is as explaining that mutual information objectives almost perform distribution matching. The caveat is that prior work omits the state entropy term $-\log \rho_{\pi_z}(s)$ which provides high reward for visiting novel states, possibly explaining why these previous works have failed to scale to complex tasks.

### B.1  ALGORITHMIC SUMMARY

We summarize the resulting procedure in Algorithm 2, which we refer to as SM4 (State Marginal Matching with Mixtures of Mixtures). The algorithm (1) fits a density model $q_z^{(m)}(s)$ to approximate the state marginal distribution for each policy $\pi_z$; (2) learns a discriminator $d^{(m)}(z \mid s)$ to predict which policy $\pi_z$ will visit state $s$; and (3) uses RL to update each policy $\pi_z$ to maximize the expected return of its corresponding reward function $r_z(s) \triangleq \log p^*(s) - \log \rho_{\pi_z}(s) + \log p(z \mid s) - \log p(z)$ derived in Equation 8.

The only difference from Algorithm 1 is that we learn a discriminator $d(z \mid s)$, in addition to updating the density models $q_z(s)$ and the policies $\pi_z(a \mid s)$. Jensen's inequality tells us that maximizing the log-density of the learned discriminator will maximize a lower bound on the true density (see Agakov (2004)):

$$\mathbb{E}_{s\sim\rho_{\pi_z}(s),z\sim p(z)}[\log d(z \mid s)] \leq \mathbb{E}_{s\sim\rho_{\pi_z}(s),z\sim p(z)}[\log p(z \mid s)]$$

---

**Algorithm 2** State Marginal Matching with Mixtures of Mixtures (SM4)

---

**Input:** Target distribution $p^*(s)$
Initialize policy $\pi_z(a \mid s)$, density model $q_z(s)$, discriminator $d(z \mid s)$, and replay buffer $\mathcal{B}$.
**while** not converged **do**
    **for** $z = 1, \cdots, n$ **do**                          $\triangleright$ (1) Update density model for each policy $\pi_z$.
        $q_z^{(m)} \leftarrow \arg\max_q \mathbb{E}_{\{s \mid (z',s) \sim \mathcal{B}^{(m-1)}, z'=z\}} [\log q(s)]$
    $d^{(m)} \leftarrow \arg\max_d \mathbb{E}_{(z,s) \sim \mathcal{B}^{(m-1)}} [\log d(z \mid s)]$           $\triangleright$ (2) Update discriminator.
    **for** $z = 1, \cdots, n$ **do**
        $r_z^{(m)}(s) \triangleq \log p^*(s) - \log q_z^{(m)}(s) + \log d^{(m)}(z \mid s) - \log p(z)$
        $\pi_z^{(m)} \leftarrow \arg\max_\pi \mathbb{E}_{\rho_\pi(s)} \left[ r_z^{(m)}(s) \right]$           $\triangleright$ (3) Update each policy $\pi_z$.
    Sample latent skill $z^{(m)} \sim p(z)$
    Sample transitions $\{(s_t, a_t, s_{t+1})\}_{t=1}^T$ with $\pi_z^{(m)}(a \mid s)$
    $\mathcal{B}^{(m)} \leftarrow \mathcal{B}^{(m-1)} \cup \{(z^{(m)}, s_t, a_t, s_{t+1})\}_{t=1}^T$
**return** $\{\{\pi_1^{(1)}, \cdots, \pi_n^{(1)}\}, \cdots, \{\pi_1^{(m)}, \cdots, \pi_n^{(m)}\}\}$

---

Algorithm 2. An algorithm for learning a *mixture* of policies $\pi_1, \pi_2, \cdots, \pi_n$ that do state marginal matching *in aggregate*. The algorithm (1) fits a density model $q_z^{(m)}(s)$ to approximate the state marginal distribution for each policy $\pi_z$; (2) learns a discriminator $d^{(m)}(z \mid s)$ to predict which policy $\pi_z$ will visit state $s$; and (3) uses RL to update each policy $\pi_z$ to maximize the expected return of its corresponding reward function derived in Equation 8: $r_z(s) \triangleq \log p^*(s) - \log \rho_{\pi_z}(s) + \log p(z \mid s) - \log p(z)$. In our implementation, the density model $q_z(s)$ is a VAE that inputs the concatenated vector $\{s, z\}$ of the state $s$ and the latent skill $z$ used to obtain this sample $s$; and the discriminator is a feedforward MLP. The algorithm returns the historical average of mixtures of policies (a total of $n \cdot m$ policies).

In our implementation, the density model $q_z(s)$ is a VAE that inputs the concatenated vector $\{s, z\}$ of the state $s$ and the latent skill $z$ used to obtain this sample $s$; and the discriminator is a feedforward MLP. The algorithm returns the historical average of mixtures of policies (a total of $n \cdot m$ policies). Our implementation uses a uniform categorical distribution for the prior $p(z)$, and does not implement the update for $p(z)$.

Note that the updates for each $z$ can be conducted in parallel. A distributed implementation would emulate broadcast-collect algorithms (Lynch, 1996), with each worker updating the policy independently, and periodically aggregating results to update the discriminator $d(z \mid s)$. Such a distributed implementation has the appealing property that each compute node would explore a different part of the state space. While there has been some work on multi-agent coordinated exploration (Parisotto et al., 2019) and concurrent exploration (Dimakopoulou & Van Roy, 2018), it remains a fairly unexplored area (pun intended) and we believe that SMM with Mixtures of Mixtures offers a simple approach to this problem.

## B.2   ABLATION STUDY

To understand the relative contribution of each component in the SM4 objective (Equation 8), we compare SM4 to baselines that lack conditional state entropy $\mathcal{H}_{\pi_z}[s] = -\log \rho_{\pi_z}(s)$, latent conditional action entropy $\log p(z \mid s)$, or both (i.e, SAC). In Figure 5a, we plot the training time performance on the *Navigation* task with 3 halls of length 50. We see that SM4 relies heavily on both key differences from SAC.

In Figure 5b, we study the effect of mixture modelling on test-time exploration in the *Manipulation* environment. After running SMM/SM4 with a uniform distribution, we count the number of episodes required to find an (unknown) goal state. We run each method for the same number of environment transitions; a mixture of three policies *does not* get to take three times more transitions. We find that increasing the number of mixture components increases the agents success. However, the effect was smaller when using historical averaging. Taken together, this result suggests that efficient exploration requires *either* historical averaging *or* mixture modelling, but might not need both.

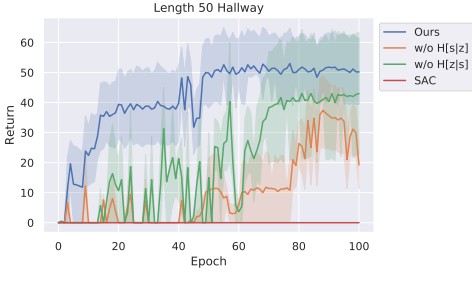
(a) Train-time Performance on *Navigation*

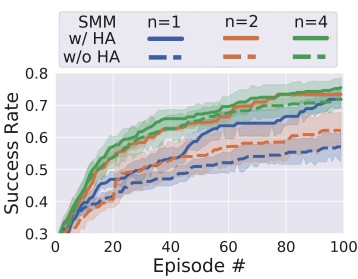
(b) Test-time Performance on *Manipulation*

Figure 5: **Ablation Analysis of State Marginal Matching with Mixtures of Mixtures (SM4)**. **(a)**: On the *Navigation* task, we compare SM4 (with three mixture components) against ablation baselines that lack conditional state entropy, latent conditional action entropy, or both (i.e., SAC) in the SM4 objective (Equation 8). We see that both terms contribute heavily to the exploration ability of SM4, but the state entropy term is especially critical. **(b)**: We compare SMM/SM4 with different numbers of mixtures, and with vs. without historical averaging. We found that increasing the number of latent mixture components $n \in \{1, 2, 4\}$ accelerates exploration, as does historical averaging. Error bars show std. dev. across 4 random seeds.

## C    CHOOSING $p^*(s)$ FOR GOAL-REACHING TASKS

In general, the choice of the target distribution $p^*(s)$ will depend on the distribution of test-time tasks. In this section, we consider the special case where the test-time tasks correspond to goal-reaching derive the optimal target distribution $p^*(s)$. We consider the setting where goals $g \sim p_g(g)$ are sampled from some known distribution. Our goal is to minimize the number of episodes required to reach that goal state. We define reaching the goal state as visiting a state that lies within an $\epsilon$ ball of the goal, where both $\epsilon > 0$ and the distance metric are known.

We start with a simple lemma that shows that the probability that we reach the goal at any state in a trajectory is at least the probability that we reach the goal at a randomly chosen state in that same trajectory. Defining the binary random variable $z_t \triangleq \mathbb{1}(\|s_t - g\| \leq \epsilon)$ as the event that the state at time $t$ reaches the goal state, we can formally state the claim as follows:

**Lemma C.1.**
$$p\left(\sum_{t=1}^{T} z_t > 0\right) \geq p(z_\mathbf{t}) \qquad where \quad \mathbf{t} \sim Unif[1, \cdots, H] \tag{9}$$

*Proof.* We start by noting the following implication:
$$z_\mathbf{t} = 1 \implies \sum_{t=1}^{T} z_t > 0 \tag{10}$$

Thus, the probability of the event on the RHS must be at least as large as the probability of the event on the LHS:
$$p(z_\mathbf{t}) \leq p\left(\sum_{t=1}^{T} z_t > 0\right) \tag{11}$$
$\square$

Next, we look at the expected number of *episodes* to reach the goal state. Since each episode is independent, the expected hitting time is simply
$$\text{HITTINGTIME}(s) = \frac{1}{p(\text{some state reaches } s)} = \frac{1}{p\left(\sum_{t=1}^{T} z_t > 0\right)} \leq \frac{1}{p(z_\mathbf{t})} \tag{12}$$

Note that we have upper-bounded the hitting time using Lemma C.1. Since the goal $g$ is a random variable, we take an expectation over $g$:
$$\mathbb{E}_{s \sim p_g(s)}\left[\text{HITTINGTIME}(s)\right] \leq \mathbb{E}_{s \sim p(s)}\left[\frac{1}{p(z_\mathbf{t})}\right] \tag{13}$$

We can rewrite the RHS using $p^*(s)$ to denote the target state marginal distribution:

$$\mathbb{E}_{s \sim p^*(s)}\left[\text{HITTINGTIME}(s)\right] \leq \mathbb{E}_{s \sim p_g(s)}\left[\frac{1}{\int p^*(\tilde{s})\mathbb{1}(\|s - \tilde{s}\| \leq \epsilon)d\tilde{s}}\right] \triangleq \mathcal{F}(p^*) \qquad (14)$$

We will minimize $\mathcal{F}$, an upper bound on the expected hitting time.

**Lemma C.2.** *The state marginal distribution $p^*(s) \propto \sqrt{\tilde{p}(s)}$ minimizes $\mathcal{F}(p^*)$, where $\tilde{p}(s) \triangleq \int p_g(\tilde{s})\mathbb{1}(\|s - \tilde{s}\| \leq \epsilon)d\tilde{s}$ is a smoothed version of the target density.*

Before presenting the proof, we provide a bit of intuition. In the case where $\epsilon \to 0$, the optimal target distribution is $p^*(s) \propto \sqrt{p_g(s)}$. For non-zero $\epsilon$, the policy in Lemma C.2 is equivalent to convolving $p_g(s)$ with a box filter before taking the square root. In both cases, we see that the optimal policy does distribution matching to some function of the goal distribution. Note that $\tilde{p}(\cdot)$ may not sum to one and therefore is not a proper probability distribution.

*Proof.* We start by forming the Lagrangian:

$$\mathcal{L}(p^*) \triangleq \int \frac{p_g(s)}{\int p^*(\tilde{s})\mathbb{1}(\|s - \tilde{s}\| \leq \epsilon)d\tilde{s}}ds + \lambda\left(\int p^*(\tilde{s})d\tilde{s} - 1\right) \qquad (15)$$

The first derivative is

$$\frac{d\mathcal{L}}{dp^*(\tilde{s})} = \int \frac{-p_g(s)\mathbb{1}(\|s - \tilde{s}\| \leq \epsilon)}{p^{*2}(\tilde{s})}ds + \lambda = 0 \qquad (16)$$

Note that the second derivative is positive, indicating that this Lagrangian is convex, so all stationary points must be global minima:

$$\frac{d^2\mathcal{L}}{dp^*(\tilde{s})^2} = \int \frac{2p_g(s)\mathbb{1}(\|s - \tilde{s}\| \leq \epsilon)}{p^{*3}(\tilde{s})}ds > 0 \qquad (17)$$

Setting the first derivative equal to zero and rearranging terms, we obtain

$$\pi(\tilde{s}) \propto \sqrt{\int p_g(s)\mathbb{1}(\|s - \tilde{s}\| \leq \epsilon)ds} \qquad (18)$$

Renaming $\tilde{s} \leftrightarrow s$, we obtain the desired result. $\qquad \square$

## C.1 CONNECTIONS TO GOAL-CONDITIONED RL

Goal-Conditioned RL (Kaelbling, 1993; Nair et al., 2018; Held et al., 2017) can be viewed as a special case of State Marginal Matching when the goal-sampling distribution is learned jointly with the policy. In particular, consider the State Marginal Matching with a mixture policy (Algorithm 2), where the mixture component $z$ maps bijectively to goal states $g$. In this case, we learn goal-conditioned policies of the form $\pi(a \mid s, g)$. We start by swapping $g$ for $z$ in the SMM objective with Mixtures of Policies (Equation 8):

$$D_{\text{KL}}(\rho_\pi(s) \parallel p^*(s)) = \mathbb{E}_{\substack{g \sim \pi(g) \\ s \sim \pi(s|g)}}\left[\log p^*(s) + \log p(g \mid s) - \log \rho_\pi(s \mid g) - \log \pi(g)\right] \qquad (19)$$

The second term $p(g \mid s)$ is an estimate of which goal the agent is trying to reach, similar to objectives in intent inference (Ziebart et al., 2009; Xie et al., 2013). The third term $\pi(s \mid g)$ is the distribution over states visited by the policy when attempting to reach goal $g$. For an optimal goal-conditioned policy in an infinite-horizon MDP, both of these terms are Dirac functions:

$$\pi(g \mid s) = \rho_\pi(s \mid g) = \mathbb{1}(s = g) \qquad (20)$$

In this setting, the State Marginal Matching objective simply says to sample goals $g \sim \pi(g)$ with probability equal to the density of that goal under the target distribution.

$$D_{\text{KL}}(\rho_\pi(s) \parallel p^*(s)) = \mathbb{E}_{\substack{g \sim \pi(g) \\ s \sim \pi(s|g)}}\left[\log p^*(s) - \log \pi(g)\right] \qquad (21)$$

Whether goal-conditioned RL is the preferable way to do distribution matching depends on (1) the difficulty of sampling goals and (2) the supervision that will be provided at test time. It is natural to use goal-conditioned RL in settings where it is easy to sample goals, such as when the space of goals is small and finite or otherwise low-dimensional. If a large collection of goals is available apriori, we could use importance sampling to generate goals to train the goal-conditioned policy (Pong et al., 2019). However, many real-world settings have high-dimensional goals, which can be challenging to sample. While goal-conditioned RL is likely the right approach when we will be given a test-time task, a latent-conditioned policy may explore better in settings where the goal-state is not provided at test-time.

## D  ADDITIONAL EXPERIMENTS & EXPERIMENTAL DETAILS

### D.1  ENVIRONMENT DETAILS

We summarize the environment parameters for Navigation (Figures 2, 5a), *Manipulation* (Figures 3, 5b, 7, 8, 9, 10), and *D'Claw* (Figure 4) in Table 1.

**Navigation**: Episodes have a maximum time horizon of 100 steps. The environment reward is

$$r_{\text{env}}(s) = \begin{cases} p_i & \text{if } \|s_{\text{robot}} - g_i\|_2^2 < \epsilon \text{ for any } i \in [n] \\ 0 & \text{otherwise} \end{cases}$$

where $s_{xy}$ is the xy-position of the agent. We used a uniform target distribution over the end of all $m$ halls, so the environment reward at training time is $r_{\text{env}}(s) = \frac{1}{m}$ if the robot is close enough to the end of any of the halls.

We used a fixed hall length of 10 in Figures 2b and 2c, and length 50 in Figure 5a. All experiments used $m = 3$ halls, except in Figure 2c where we varied the number of halls $\{3, 5, 7\}$.

**Manipulation**. We used the simulated Fetch Robotics arm[5] implemented by Plappert et al. (2018) using the MuJoCo simulator Todorov et al. (2012). The state vector $s \in \mathbb{R}^{28}$ includes the xyz-coordinates $s_{\text{obj}}, s_{\text{robot}} \in \mathbb{R}^3$ of the block and the robot gripper respectively, as well as their velocities, orientations, and relative position $s_{\text{obj}} - s_{\text{robot}}$. At the beginning of each episode, we spawn the object at the center of the table, and the robot gripper above the initial block position. We terminate each episode after 50 environment steps, or if the block falls off the table.

We considered two target state marginal distributions. In *Manipulation-Uniform*, the target density is given by

$$p^*(s) \propto \exp\left(\alpha_1 r_{\text{goal}}(s) + \alpha_2 r_{\text{robot}}(s) + \alpha_3 r_{\text{action}}(s)\right)$$

where $\alpha_1, \alpha_2, \alpha_3 > 0$ are fixed weights, and the rewards

$$r_{\text{goal}}(s) := 1 - \mathbb{1}(s_{\text{obj}} \text{ is on the table surface})$$
$$r_{\text{robot}}(s) := \mathbb{1}(\|s_{\text{obj}} - s_{\text{robot}}\|_2^2 < 0.1)$$
$$r_{\text{action}}(s) := -\|a\|_2^2$$

correspond to (1) a uniform distribution of the block position over the table surface (the agent receives +0 reward while the block is on the table), (2) an indicator reward for moving the robot gripper close to the block, and (3) action penalty, respectively. The environment reward is a weighted sum of the three reward terms: $r_{\text{env}}(s) \triangleq 20 r_{\text{goal}}(s) + r_{\text{robot}}(s) + 0.1 r_{\text{action}}(s)$. At test-time, we sample a goal block location $g \in \mathbb{R}^3$ uniformly on the table surface, and the goal is not observed by the agent.

In *Manipulation-Half*, the target state density places higher probability mass to states where the block is on the left-side of the table. This is implemented by replacing $r_{\text{goal}}(s)$ with a reward function that gives a slightly higher reward +0.1 for states where the block is on the left-side of the table.

**D'Claw**. The *D'Claw* robot (Ahn et al., 2019; Zhu et al., 2019)[6] controls three claws to rotate a valve object. The environment consists of a 9-dimensional action space (three joints per claw) and a 12-dimensional observation space that encodes the joint angles and object orientation. We fixed each

---

[5] https://fetchrobotics.com/
[6] www.roboticsbenchmarks.org

Table 1: **Environment parameters** specifying the observation space dimension $|\mathcal{S}|$; action space dimension $|\mathcal{A}|$; max episode length $T$; the environment reward, related to the target distribution by $\exp\{r_{\text{env}}(s)\} \propto p^*(s)$, and other environment parameters.

| Environment | $|\mathcal{S}|$ | $|\mathcal{A}|$ | $T$ | Env Reward ($\log p^*(s)$) | Other Parameters | Figure |
|---|---|---|---|---|---|---|
| *Navigation* | 2 | 2 | 100 | Uniform over all $m$ halls | # Halls: 3, 5, 7 Hall length: 10 | 2 |
| | | | | Uniform over all $m$ halls | # Halls: 3 Hall length: 50 | 5a |
| *Manipulation* | 25 | 4 | 50 | Uniform block pos. over table surface | | 3b, 3c, 5b |
| | | | | More block pos. density on left-half of table | | 3d |
| *D'Claw* | 12 | 9 | 50 | Uniform object angle over $[-180°, 180°]$ | | 4 |

episode at 50 timesteps, which is about 5 seconds on the real robot. In the hardware experiments, each algorithm was trained on the same four *D'Claw* robots to ensure consistency.

We defined the target state distribution to place uniform probability mass over all object angles in $[-180°, 180°]$. It also incorporates reward shaping terms that place lower probability mass on states with high joint velocity and on states with joint positions that deviate far from the initial position (see (Zhu et al., 2019)).

## D.2 GAIL

GAIL assumes access to expert demonstrations, which SMM and the other exploration methods do not require. To compare GAIL with the exploration methods on a level footing, we sampled synthetic states from $p^*(s)$ to train GAIL, and restricted the GAIL discriminator input to states only (no actions).

For *D'Claw* (Fig. 4), we sampled the valve object angle uniformly in $[-180°, 180°]$. For *Manipulation-Uniform* (Fig. 3c), we sampled object positions $s_{\text{object}}$ uniformly on the table surface, and tried two different sampling distributions for the gripper position $s_{\text{robot}}$ (see Fig. 6). For both environments, all other state dimensions were sampled uniformly in $[-10, 10]$, and used 1e4 synthetic state samples to train GAIL.

Since the state samples from $p^*(s)$ may not be reachable from the initial state, the policy may not be able to fool the discriminator. To get around this problem, we also tried training GAIL with the discriminator input restricted to only the state dimensions corresponding to the object position or gripper position (*Manipulation*), or the object angle (*D'Claw*). We summarize these GAIL ablation experiments in Fig. 6. In our experiments, we used the best GAIL ablation model to compare against the exploration baselines in Figures 3c and 4.

## D.3 VAE DENSITY MODEL

In our SMM implementation, we estimated the density of data $x$ as $p(x) \approx \text{decoder}(\hat{x} = x|z = \text{encoder}(x))$. That is, we encoded $x$ to $z$, reconstruction $\hat{x}$ from $z$, and then took the likelihood of the true data $x$ under a unit-variance Gaussian distribution centered at the reconstructed $\hat{x}$. The log-likelihood is therefore given by the mean-squared error between the data $x$ and the reconstruction $\hat{x}$, plus a constant that is independent of $x$: $\log q(x) = \frac{1}{2}\|x - \hat{x}\|_2^2 + C$.

## D.4 COMPUTATIONAL COMPLEXITY

We compare the wall-clock time of each exploration method in Table 2. The computational cost of our method is comparable with prior work.

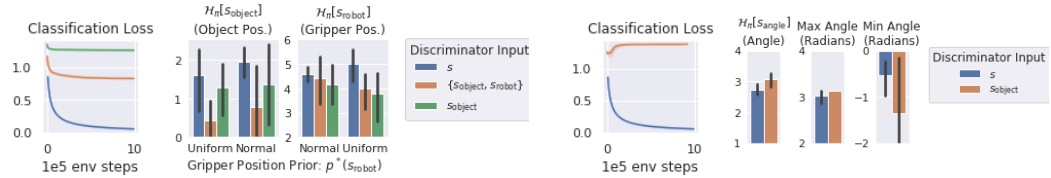

(a) GAIL ablations on *Manipulation*       (b) GAIL ablations on *D'Claw*

Figure 6: **GAIL Ablation Study**: We studied the effect of restricting the GAIL discriminator input to fewer state dimensions. **(a)** *Manipulation*: We trained the GAIL discriminator on the entire state vector $s$; on the object and gripper positions $\{s_{\text{object}}, s_{\text{robot}}\}$ only; or on the object position $s_{\text{object}}$ only. We also varied the sampling distribution for the gripper position, $p^*(s_{\text{robot}})$: we compare using a normal distribution, $\mathcal{N}(s_{\text{object}}, I_3)$, to sample gripper positions closer to the object, versus a uniform distribution, Uniform$[-10, 10]$, for greater entropy of the sampled gripper positions. We observe that sampling gripper positions closer to the object position improves the entropy of the object position $\mathcal{H}_\pi[s_{\text{object}}]$, but hurts the entropy of the gripper position $\mathcal{H}_\pi[s_{\text{robot}}]$. **(b)** *D'Claw*: We restricted the discriminator to the entire state vector $s$, or to the object angle and position $s_{\text{object}}$. **Analysis**: In both domains, we observe that restricting the discriminator input to fewer state dimensions (e.g., to $s_{\text{object}}$) makes the discriminator less capable of distinguishing between expert and policy states (orange and green curves). On the other hand, training on the entire state vector $s$ causes the discriminator loss to approach 0 (i.e., perfect classification), partly because some of the "expert" states sampled from $p^*(s)$ are not reachable from the initial state, and the policy is thus unable to fool the discriminator.

Table 2: Per-epoch wall-clock time on the *Manipulation* environment. One epoch is 1e3 env. steps.

| SAC | ICM | Count | **SMM (ours)** | PseudoCounts |
|---|---|---|---|---|
| 17.95s (+0%) | 22.74s (+27%) | 25.24s (+41%) | **25.82s (+44%)** | 33.87s (+89%) |

### D.5 ALGORITHM HYPERPARAMETERS

We summarize hyperparameter settings in Table 3. All algorithms were trained for 1e5 steps on *Navigation*, 1e6 steps on *Manipulation*, 1e6 steps on *D'Claw* Sim2Real, and 1e5 steps on *D'Claw* hardware.

**Loss Hyperparameters**. For each exploration method, we tuned the weights of the different loss components. *SAC reward scale* controls the weight of the action entropy reward relative to the extrinsic reward. *Count coeff* controls the intrinsic count-based exploration reward w.r.t. the extrinsic reward and SAC action entropy reward. Similarly, *Pseudocount coeff* controls the intrinsic pseudocount exploration reward. *SMM coeff for $\mathcal{H}[s \mid z]$ and $\mathcal{H}[z \mid s]$* control the weight of the different loss components (state entropy and latent conditional entropy) of the SMM objective in Equation 8.

**Historical Averaging**. In the *Manipulation* experiments, we tried the following sampling strategies for historical averaging: (1) *Uniform*: Sample policies uniformly across training iterations. (2) *Exponential*: Sample policies, with recent policies sampled exponentially more than earlier ones. (3) *Last*: Sample the $N$ latest policies uniformly at random. We found that *Uniform* worked less well, possibly due to the policies at early iterations not being trained enough. We found negligible difference in the state entropy metric between *Exponential* vs. *Last*, and between sampling 5 vs. 10 historical policies, and we also note that it is unnecessary to keep checkpoints from every iteration.

**Network Hyperparameters**. For all algorithms, we use a Gaussian policy with two hidden layers with Tanh activation and a final fully-connected layer. The Value function and Q-function each are a feedforward MLP with two hidden layers with ReLU activation and a final fully-connected layer. Each hidden layer is of size 300 (SMM, SAC, ICM, C, PC) or 256 (GAIL). The same network configuration is used for the SMM discriminator, $d(z \mid s)$, and the GAIL discriminator, but with different input and output sizes. The SMM density model, $q(s)$, is modeled by a VAE with encoder and decoder networks each consisting of two hidden layers of size (150, 150) with ReLU activation. The same VAE network configuration is used for Pseudocount.

**GAIL Hyperparameters**: The replay buffer is filled with 1e4 random actions before training, for training stability. We perform one discriminator update per SAC update. For both *Manipulation* and *D'Claw*, we used 1e4 states sampled from $p^*(s)$. Other hyperparameter settings, such as batch size for both discriminator and policy updates, are summarized in Table 3. We observed that GAIL training is more unstable compared to the exploration baselines. Thus, for GAIL, we did not take the final iterate (e.g., policy at convergence) but instead used early termination (e.g., take the best iterate according to the state entropy metric).

### D.6 VISUALIZING THE MANIPULATION ENVIRONMENT

We visualize where different methods push the block in the *Manipulation* environment. More precisely, we visualize the log state marginal $\log \rho_{\pi_z}(s)$ over block XY-coordinates $s = (x, y)$ in Figures 7 and 8. In Figure 9, we plot goals sampled at test-time, colored by the number of episodes each method required to push the block to that goal location. Blue dots indicate that the agent found the goal quickly. We observe that SMM has the most blue dots, indicating that it succeeds in exploring a wide range of states at test-time.

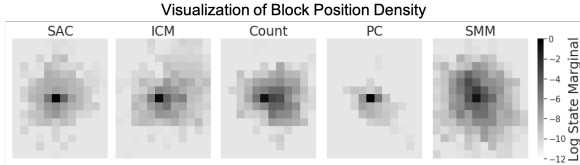

Figure 7: The log state marginal $\log \rho_\pi(s)$ over block XY-coordinates, averaged over 1e3 epochs.

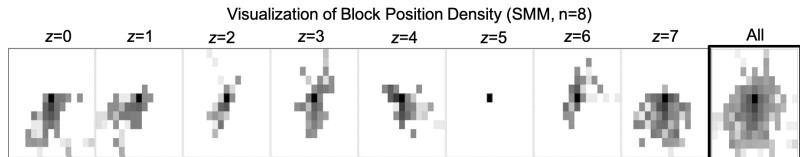

Figure 8: **SM4 with Eight Mixture Components**. The log state marginal $\log \rho_{\pi_z}(s)$ over block XY-coordinates for each latent skill $z \in \{0, \ldots, 7\}$, averaged over 1000 epochs.

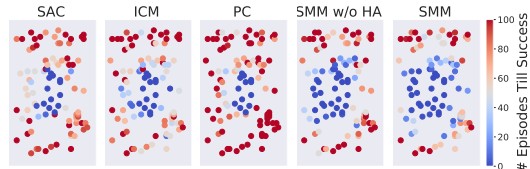

Figure 9: Goals sampled uniformly on the table surface, colored by the number of episodes until the policy finds the goal. Red (100 episodes) indicates failure. The block always starts at the center.

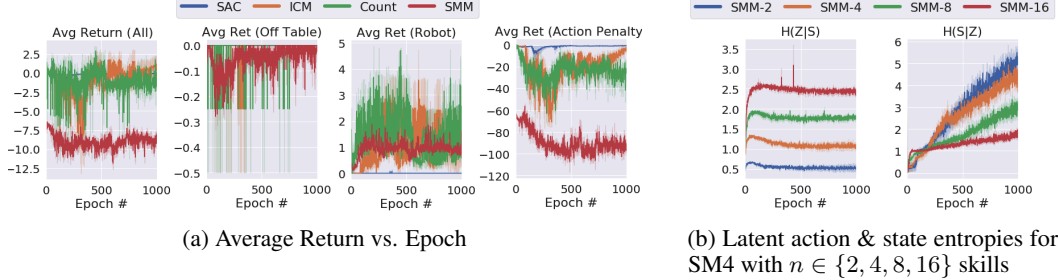

(a) Average Return vs. Epoch

(b) Latent action & state entropies for SM4 with $n \in \{2, 4, 8, 16\}$ skills

Figure 10: **Train curves on *Manipulation***. One epoch is 1e3 steps. **(a)** . The environment reward is a weighted sum of three terms: $r_{\text{goal}}(s)$ (+0 if object is on table, -1 otherwise), $r_{\text{robot}}(s)$ (+1 if robot gripper is close to block), and $r_{\text{action}}$ (action penalty term), with weights -20, 1, 0.1 respectively (see Appendix D.1). The three exploration methods (ICM, Count, SMM) also optimize an auxilliary exploration loss, which makes the agent more likely to move around the block. Compared to SAC, this causes the exploration methods to get worse returns for $r_{\text{goal}}(s)$ and $r_{\text{action}}(s)$ (due to the agent moving the block around), but also quickly learns to maximize the sparse reward $r_{\text{robot}}(s)$ (indicator reward for moving gripper within a threshold distance to the block). **(b)** The latent action entropy $\mathcal{H}[z \mid s]$ (discriminator) and latent state entropy $\mathcal{H}[s \mid z]$ (density model) per epoch.

Table 3: **Hyperparameter Settings**. Hyperparameters were chosen according to the following eval metrics: *Manip.-Uniform*: State entropy of the discretized gripper and block positions (bin size 0.05), after rolling out the trained policy for 50K env steps. *Manip.-Half*: $D_{\mathrm{KL}}(p^*(s) \,\|\, \rho_\pi(s))$ and $\mathrm{TV}(p^*(s), \rho_\pi(s))$ of the discretized gripper and block positions (bin size 0.01), after rolling out the trained policy for 50K env steps. *2D Navigation*: State entropy of the discretized XY-positions of the trained policy. *D'Claw*: State entropy of the object angle.

| Environment | Algorithm | Hyperparameters Used | Hyperparameters Considered |
|---|---|---|---|
| All | SMM, SAC, ICM, Count, Pseudocount | Batch size: 128
1e6 env training steps
RL discount: 0.99
Network size: 300
Policy lr: 3e-4
Q-function lr: 3e-4
Value function lr: 3e-4 | N/A (Default SAC hyperparameters) |
| | GAIL | 1e6 env training steps
Policy lr: 1e-5
Critic lr: 1e-3
# Random actions
   before training: 1e4
Network size: 256 | N/A (Default GAIL hyperparameters) |
| *Navigation* (Fig. 2, 5a) | SMM, SAC | SAC reward scale: 25 | SAC reward scale: 1e-2, 0.1, 1, 10, 25, 100 |
| | SMM | SMM $\mathcal{H}[s \mid z]$ coeff: 1
SMM $\mathcal{H}[z \mid s]$ coeff: 1 | SMM $\mathcal{H}[s \mid z]$ coeff: 1e-3, 1e-2, 1e-1, 1, 10
SMM $\mathcal{H}[z \mid s]$ coeff: 1e-3, 1e-2, 1e-1, 1, 10 |
| *Manip.-Uniform* (Fig. 3b, 3c, 5b) | SMM | Num skills: 4
VAE lr: 1e-2
SMM $\mathcal{H}[s \mid z]$ coeff: 1
SMM $\mathcal{H}[z \mid s]$ coeff: 1
HA sampling: Exponential
# HA policies: 10
SMM Latent Prior Coeff: 1 | Num skills: 1, 2, 4, 8, 16
VAE lr: 1e-4, 1e-3, 1e-2


HA sampling: Exponential, Uniform, Last
# HA policies: 5, 10
SMM Latent Prior Coeff: 1, 4 |
| | SAC | SAC reward scale: 0.1 | SAC reward scale: 0.1, 1, 10, 100 |
| | Count | Count coeff: 10
Histogram bin width: 0.05 | Count coeff: 0.1, 1, 10 |
| | Pseudocount | Pseudocount coeff: 1
VAE lr: 1e-2 | Pseudocount coeff: 0.1, 1, 10
(Use same VAE lr as SMM) |
| | ICM | Learning rate: 1e-3 | Learning rate: 1e-4, 1e-3, 1e-2 |
| | GAIL | Batch size: 512
# SAC updates per step: 1
Discriminator input: $s$
Training iterate: 1e6
# State Samples: 1e4 | Batch size: 128, 512, 1024
# SAC updates per step: 1, 4
Discriminator input: $s$, $s_{\mathrm{object}}$, $\{s_{\mathrm{object}}, s_{\mathrm{robot}}\}$
Training iterate: 1e5, 2e5, 3e5, ..., 9e5, 1e6
# State Samples: 1e4 |
| *Manip.-Half* (Fig. 3d) | SMM, SAC, ICM, Count | SAC reward scale: 0.1 | (Best reward scale for Manip.-Uniform) |
| | SMM | Num skills: 4
SMM $\mathcal{H}[s \mid z]$ coeff: 1
SMM $\mathcal{H}[z \mid s]$ coeff: 1 | Num skills: 1, 2, 4, 8 |
| | Count | Count coeff: 10
Histogram bin width: 0.05 | Count coeff: 0.1, 1, 10 |
| | ICM | Learning rate: 1e-3 | Learning rate: 1e-4, 1e-3, 1e-2 |
| *D'Claw* (Fig. 4) | SMM, SAC | SAC reward scale: 5 | SAC reward scale: 1e-2, 0.1, 1, 5, 10, 100 |
| | SMM | SMM $\mathcal{H}[s \mid z]$ coeff: 250 | SMM $\mathcal{H}[s \mid z]$ coeff: 1, 10, 100, 250, 500, 1e3 |
| | Count | Count coeff: 10
Histogram bin width: 0.05 | Count coeff: 1, 10
Histogram bin width: 0.05, 0.1 |
| | Pseudocount | Pseudocount coeff: 1
VAE lr: 1e-3 | Pseudocount coeff: 1, 10
VAE lr: 1e-1, 1e-2, 1e-3 |
| | ICM | Learning rate: 1e-3
VAE lr: 1e-1 | Learning rate: 1e-2, 1e-3, 1e-4
VAE lr: 1e-1, 1e-2, 1e-3 |
| | GAIL | Batch size: 512
# SAC updates per step: 4
Discriminator input: $s_{\mathrm{object}}$
Training iterate: 1e5
# State Samples: 1e4 | Batch size: 128, 512, 1024
# SAC updates per step: 1, 4
Discriminator input: $s$, $s_{\mathrm{object}}$
Training iterate: 1e5, 2e5, 3e5, ..., 9e5, 1e6
# State Samples: 1e4 |

