# OpenReview forum: "Efficient Exploration via State Marginal Matching"
_ICLR.cc/2020/Conference — Reject_

### Official Review · AnonReviewer2 · 2019-10-22
**Official Blind Review #2**

**Rating:** 3

**Review:**

Update: I thank the authors for their response and I think the added baselines and more in depth discussion of prior work have improved the paper. However, given the limited novelty of the technical contribution, I believe the experimental section should be further extended (i.e. add a wider variety of domains and make thorough comparisons to relevant methods) in order to draw more general and robust conclusions.

Summary:

This paper proposes tackles the problem of exploration in RL, with a focus on learning an exploration policy that can be used for a variety of different tasks. They introduce a formal exploration objective that promotes efficient exploration and provides a mechanism for injecting prior knowledge about the task. They also design a practical algorithm to maximize the exploration objective. Finally, they empirically show that  their method can outperform other SOTA exploration methods on challenging exploration tasks for locomotion and manipulation,  both in simulation and in the real world.

Main Comments:

I’ve found the mathematical formulation to be sound and the empirical evaluation convincing. Overall, the paper is clearly written and the authors are quite transparent about the assumptions made. In addition, the problem of learning exploration strategies that are task-agnostic and force the agent to effectively explore within each episode (since the goal or task is not observed) is an important problem for RL and perhaps a more realistic setting than the single fixed-task one. However, I believe some important methodological details are missing from the paper and the empirical evaluation can be improved. In particular, the paper would be more convincing if it contained comparisons against SOTA exploration (e.g. curiosity-driven, pseudo-counts, noisy-networks etc.) and inverse reinforcement learning (e.g. GAIL) methods for all the environments. Such baselines are completely missing in the Navigation and Real-World tasks.

However, as the authors note, most baselines used for comparison have been designed specifically to learn from sparse rewards in single task settings and do not have any direct mechanisms for including priors or learning to explore well for any task from some distribution. So I wonder if it’d make sense to include baselines that do make use of prior knowledge such as IRL (i.e. GAIL) or some other state-matching approach. Those could be more appropriate and powerful baselines.

Another potential disadvantage of this method seems to be the need for a prior, which may be difficult to design or even lead to suboptimal policies if it is not well designed. However, as the authors note, it is still a weaker requirement than having access to demonstrations for example and the  prior could potentially be learned from human preferences / restricted feedback.

Other Comments  /  Questions:

1. SMM uses prior information about the task in the form of the target distribution. Given this, I am worried that the baselines have a clear disadvantage. Did you do anything to provide the baselines with the same type of prior knowledge about the task? It would be useful to see how they would compare if they had access to the task prior (in some way) as well.

2. Can you provide more insights into how this differs from variants of MaxEntRL and InvRL? Both analytically and in practice. I believe a more extended discussion of this would be valuable for readers and would alleviate some of the concerns regarding the novelty of this contribution and how its place in the broader literature.

3. In the Navigation environment, how would the  results change if the goal were visible  (i.e. part of the agent’s observation)? I believe that most baselines would consider that scenario and it would  be interesting to see whether the qualitative conclusions hold or not in that case. I would expect other exploration methods to  be faster in that case.

4. I also wonder if perhaps the reward is actually not that sparse  in some of these tasks but  because it is not visible, it makes the problem much harder for the baselines, which were designed to deal with very sparse reward. Can you comment on the reward sparsity in these tasks?

5. At the top of page 2, you mention that there is a training phase in which the agents learn to optimize the exploration objective and at test time, it is trained with extrinsic reward. Can you please clarify on how these stages reflect in the results and what is  the regime used for the other baselines? Are they also pretrained on a variety of tasks with only their exploration bonus / intrinsic reward and then fine-tuned with extrinsic reward?

6. In Figure 2 (c), why is it that the gap between SMM and SAC decreases as the  number  of halls increases? This seems counterintuitive and I would’ve expected to increase since I do not see why SAC would get better and SMM would get worse.

7. How do you learn the density model? You mention the use of a VAE but the details of how this is trained are not specified.

8. On page 5 before section 4,  you mention that you approximate the historical average  of the density model with the most recent iterate. Can you include ablations on how good this approximation is and how the results change  if you were using the historical average instead?

9. In Figure 4 (b), SMM’s variance of the positive value of the angle differs significantly from the negative one. This strikes me as counterintuitive. Do you have any  intuition on why that is?








**Experience Assessment:**

I have read many papers in this area.

**Review Assessment: Checking Correctness Of Derivations And Theory:**

I assessed the sensibility of the derivations and theory.

**Review Assessment: Checking Correctness Of Experiments:**

I assessed the sensibility of the experiments.

**Review Assessment: Thoroughness In Paper Reading:**

I read the paper at least twice and used my best judgement in assessing the paper.

---

> ### Author Response · Authors · 2019-11-15
> **Author Response**
>
> Thank you for your detailed response. Per your suggestion, we have added a comparison to state-of-the-art exploration baselines on the real-world manipulation task (Fig 4b). We have also added a comparison with GAIL on both the simulated and real-world manipulation experiments (Fig 3c, 4b). GAIL makes slightly different assumptions than State Marginal Matching, and we explain how we compare the methods on a level footing in Appendix D.2. In both cases, State Marginal Matching outperforms all baselines, including GAIL.
>
> We address your questions below:
>
> 1. There might be a misunderstanding here: the exploration baselines have access to exactly the same information as SMM. The only supervision given to both is the environment reward (see Appendix D.1). SMM interprets this reward as the log probability of a target distribution.
>
> 2. MaxEnt RL is maximizing entropy in action space. As shown in Figures 2 and 4, exploration in state space, as done by SMM, is substantially more effective. While SMM is taking a target state distribution and returning a policy, inverse RL is doing the opposite, taking an expert policy and returning a distribution over trajectories.
>
> 3. You are correct that, in the navigation setting, it is more realistic to consider the setting where goals are observed by the agent. The aim of the navigation setting we constructed was not to be as realistic as possible, but rather to create a testbed for exploration where we can parametrically vary the difficulty of exploration. The key property of hard exploration tasks is that states with high reward (i.e., the goal) is not known apriori, so the agent must explore to find these high-reward states. The navigation environment is designed to mimic these dynamics.
>
> 4. The navigation task used a purely sparse reward, while the simulated and real-world manipulation environments used a combination of dense and sparse rewards (see Appendix D.1). All of these tasks used in our experiments are challenging precisely because of the sparse components of their rewards.
>
> 5. We will clarify when in the training process each of the figures was created. Fig 2c was created by just considering the goals visited during the initial exploration phase. Fig 3b shows performance during the task-specific adaptation phase, after all task-agnostic exploration had taken place. Fig 3c and 3d show performance after the task-agnostic exploration phase, but before any task-specific adaptation. Fig 4b shows performance after the task-agnostic exploration phase, but before any task-specific adaptation. Fig 4c shows performance during the task-agnostic exploration phase.
>
> 6. We likewise expect SAC to get worse as the number of hallways increases. Since all of the SAC numbers are within 2 standard deviations, we attribute the slight rise to random fluctuations.
>
> 7. We estimated the density of data x as $p(x) \approx decoder(\hat{x} = x | z=encoder(x))$. That is, we encoded x to z, reconstruction $\hat{x}$ from z, and then took the likelihood of the true data x under a Gaussian distribution centered at the reconstructed $\hat{x}$. We have clarified this in Appendix D.2.
>
> 8. In Appendix D.3 ("Historical Averaging") we discussed how the results change if we sampled checkpoints uniformly vs. if we sampled later checkpoints more frequently. We found that uniform sampling worked less well, possibly due to the policies at early iterations not being trained enough.
>
> 9. As shown in Fig 4a, in the initial state, the fingers were not placed in the middle of the knob spaces, but rather were close to the knob on one side than the knob on the other side. We attribute the difference in variance to this asymmetry in the initial position.

---

### Official Review · AnonReviewer1 · 2019-10-22
**Official Blind Review #1**

**Rating:** 3

**Review:**

Summary:
The paper proposes to frame exploration in reinforcement learning as a distribution matching problem. More specifically, the proposed method (SMM) aims to minimize the reverse KL between the state distribution induced by the policy and a desired state distribution. The desired state distribution can be used to guide exploration (e.g. by penalizing bad states) or can be chosen uniformly (resulting in a policy that maximizes state entropy).
SMM iteratively optimizes the policy and a model for its induced state-distribution. The latter is used for approximating the policies state entropy. The algorithm is framed as a two-player zero-sum game between a "policy player" and a "density player". In order to justify that one of the players is assumed fixed while optimizing the other player, SMM optimizes the players against a historical average of the opponent. Such approach is known as fictitious play in game theory and ensures convergence to a Nash equilibrium.
The density player maximizes the likelihood of the states encountered during all previous roll-outs using a VAE. The policy update is framed as a standard reinforcement learning problem where the log-pdf of the target distribution serves as reward and the log-pdf of the model acts as cost. For practical reasons, the policy update does not consider an historical average but only uses the most recent density model. As the learned exploratory policy should correspond to an historical average exploration for the downstream task is achieved by sampling one of the learned policy at the beginning of each roll-out. The historical averaging also benefits prior exploration methods.
Notably, the appendix also describes a modification that learns a mixture of policies in each iteration where a discriminator approximates the responsibilities for each component which is used as additional reward to divide the search space among the components. SMM is compared to prior exploration methods, ICM and Pseudo-Counts and standard SAC, on a simulated robotic box-pushing task and against SAC on simulated point-mass problem and a real-robot valve turning task.

Significance:
Efficient exploration is arguably the main challenge of reinforcement learning as providing shaped reward functions is difficult even for experts and may lead to undesired behavior. I think that maximizing the state-entropy (or matching target state-distributions if available) is a sound objective for learning exploratory policies and the paper could, thus, be of relatively broad interest.

Novelty:
Maximizing the state entropy for exploration has already been used by (Hazan et al., 2018, reference from manuscript). However, in contrast to this prior work, SSM does not need to learn oracles that predict optimal state-distributions/policies for any given policies/reward functions. While distribution matching is a common approach to imitation learning, it has been little employed for manually specified distributions. Still, a similar objective has been used to replace reward functions by desired distributions in a RL-like setting [1] (not for exploration). However, their approach is quite restricted by assuming Gaussian target distributions.

Soundness:
If I understand correctly, fictitious play assumes optimal responses to understand that the state distribution would often be more important than provably converge to a Nash equilibrium. The paper fails to provide stopping criteria for the optimization steps of the individual players; however, I assume that only few gradient steps are used for practical reasons. Hence, I am not sure whether the actual algorithm can be justified by the theory.

The paper mentions that VAEs are used to model the state distribution. Given that VAEs are not likelihood-based, I do not understand how the reward term log(q(s)) can be computed.

Clarity:
The paper is well-written and the structure seems fine.
I think that the density estimation should be better discussed. Models of the state distribution of the policy are often highly desirable, not only for optimizing its entropy, but also, for example, for importance sampling. However, modeling the state distribution is also inherently difficult--especially for large state spaces.

Experiments:
I like that the paper uses a real robot experiment and the ablations with respect to the historical averaging are interesting. Unfortunately, the paper only compares to standard SAC on the hallway task and on the real robot task. I would not consider the entropy regularization of SAC a proper baseline for a proposed exploration technique.

Questions:
Is the same extrinsic reward, i.e. log(p*), also provided to the other exploration methods?

Some imitation learning methods such as GAIL are also able to match desired state-distributions. I think that these methods could be in principle also applied to the setting considered in the manuscript by using samples from the desired distribution as demonstrations. The paper briefly mentions that IRL methods are not applicable because they require access to trajectories, however, the discriminator of GAIL is only trained on individual state samples. I also do not see a problem of providing unachievable demonstrations to such imitation learning methods because, such like SSḾM, they would try to minimize the divergence. I think that GAIL would actually be an important baseline for the proposed method.

How does the method scale with the dimensions of the state? SSM has only be evaluated on relatively low-dimensional problems (compared to some rllab/mujoco tasks with >100 states). I would assume that obtaining meaningful density estimates in such settings might be problematic. May imitation learning methods based on discriminators actually be more promising?

SMM only considers matching state distributions. If I understand correctly, the approach could be easily generalized to state-action distributions, correct? Wouldn't it make sense for some tasks to also explore in the action space?

Decision:
I like the paper in general. The optimization problem seems well-motivated and the algorithm seems reasonable fine. I also like that the paper includes a real robot experiment. However, I am not convinced by the derivation of SMM based on fictitous play and I think that it should be better evaluated with respect to existing exploration methods and also with respect to imitation learning methods. I am therefore slightly leaning towards rejection, currently.

Typo:
"Note[sic] only does SMM explore a wider range of angles than SAC, but its ability to explore increasing[sic] throughout training, suggesting that the SMM objective is correlated with real-world metrics of exploration."

[1] Arenz, Oleg, Hany Abdulsamad, and Gerhard Neumann. "Optimal control and inverse optimal control by distribution matching." 2016 IEEE/RSJ International Conference on Intelligent Robots and Systems (IROS). IEEE, 2016.

**Experience Assessment:**

I have read many papers in this area.

**Review Assessment: Checking Correctness Of Derivations And Theory:**

I assessed the sensibility of the derivations and theory.

**Review Assessment: Checking Correctness Of Experiments:**

I assessed the sensibility of the experiments.

**Review Assessment: Thoroughness In Paper Reading:**

I read the paper at least twice and used my best judgement in assessing the paper.

---

> ### Author Response · Authors · 2019-11-15
> **Author Response**
>
> Thanks for the detailed review and suggestions for improvement. We believe that we have already included the comparison to state-of-the-art exploration methods for simulated manipulation (Figure 3). We have also incorporated your suggestion of including the exploration baselines on another task (real-world manipulation in Figure 4b), and including GAIL comparisons to both simulated and real-world manipulation results (Figures 3c, 4b), as noted in the "Revisions Summary" comment above. We also think there may be a misunderstanding about the connection with imitation learning, which we will discuss below.
>
> Imitation Learning: We think there might be a misunderstanding regarding the relationship between State Marginal Matching and imitation learning. You are correct in noting that both State Marginal Matching and many imitation learning methods (e.g., GAIL, AIRL) maximize the same objective (a KL divergence between state marginals). However, these methods have different requirements: State Marginal Matching (and other exploration methods) require a reward function, while imitation learning methods require expert trajectories. It is unclear to use how a fair comparison could be done, though we would welcome any suggestions. We have added GAIL results for Manipulation (Fig 3b, 3c) and D'Claw (Fig 4b), and also included an ablation study of GAIL (Fig 6) to explain the different variations of GAIL that we've tried.
>
> Moreover, this connection between exploration and distribution matching is, in fact, a significant part of our contribution: prior exploration methods do perform approximate distribution matching. More precisely, we show in Section 4 that a large class of exploration bonuses (those based on prediction error) are all maximizing the same objective: marginal state entropy. Our work makes this connection explicit, and explains how state distribution matching can be performed properly. This observation is useful precisely because many of the underlying ingredients, such as adversarial games and density estimation, have seen recent progress and therefore might be adopted to improve exploration methods.
>
> Other Clarifications to the Reviewer's Questions:
>
> 1. VAEs: We estimated the density of data x as $p(x) \approx decoder(\hat{x} = x | z=encoder(x))$. That is, we encoded x to z, reconstruction $\hat{x}$ from z, and then took the likelihood of the true data x under a Gaussian distribution centered at the reconstructed $\hat{x}$. We have clarified this in Appendix D.2
>
> 2. Extrinsic Reward: Yes, all exploration methods have access to exactly the same information and the same extrinsic reward (see Appendix D.1 for details). SMM interprets this reward as the log probability of a target distribution.
>
> 3. Yes, density estimation is challenging, though it continues to lie as a foundation for many exploration methods (e.g., pseudo-counts). By showing that exploration is really a problem of density estimation, our work allows progress on density estimation to be made useful for exploration.
>
> 4. State-Action Entropy: In fact, our approach does maximize the state-action entropy. The state-action entropy factors as H[a|s] + H[s]. The first term, H[a|s], is maximized by MaxEnt RL methods (e.g., SAC), while the latter is maximized with the fictitious play procedure.

---

> > ### Comment · AnonReviewer1 · 2019-11-15
> > **Thanks for the revision**
> >
> > Thank you for your reply and the additional experiments.
> >
> > 1) Unfortunately you did not comment on my notes regarding the soundness of the derivation based on fictitious play.
> >
> > 2) Regarding the relation between some imitation learning approaches and state distribution matching. I do understand that imitation learning assumes access to samples from a distribution whereas SMM assumes access to a desired distribution (potentially in the form of a reward function). However, apart from this, I do not see any major difference compared to the setting of some imitation learning algorithms.
> > Note that these methods (GAIL, AIRL, etc.) are derived for minimizing a divergence and do not necessarily assume that a divergence of zero is achievable. AIRL even minimizes the (approximately) same divergence as SSM. I believe that evaluating these methods on the SMM-scenario is straightforward by (approximately) sampling from the desired distribution. I suggested a similar experiment to the one that you added. However, I would have used a sampler to obtain samples from p* so that both methods approximately optimize the same target distribution (or reward function). Of course, using an off-the-shelf sampler to obtain samples would lead to additional computational overhead. However, depending on the reward function and the required sample accuracy the overhead might be negligible and you do not evaluate w.r.t. computational time anyway. Your additional experiments also do not seem to provide the number of demonstrations that you provided to GAIL (when using synthetic demonstrations we can and should use a large number of samples to avoid overfitting).
> >
> > 3) Regarding 4. I understand that you can maximize state-action entropy. My Question was:
> > "SMM only considers matching state distributions. If I understand correctly, the approach could be easily generalized to state-action distributions, correct? Wouldn't it make sense for some tasks to also explore in the action space?"

---

> > > ### Author Response · Authors · 2019-11-15
> > > **Author Response**
> > >
> > > Thanks for reading the response!
> > >
> > > 1) *Soundness of fictitious play*: Our convergence result assumes that the optimization for each player be solved exactly at each step. In our implementation, we approximated this by taking one gradient step at each time step. Note that this is the same approximation made in the GAN literature. For example, the analysis in Section 4 of [Goodfellow 2014] assumes that the discriminator is optimal.
> > >
> > > 2) *Comparison with IRL*: State Marginal Matching does not require that there exist a policy that can exactly match the target distribution. We do assume (Assumption 1) that the _density_ model is fit exactly to the policy. We emphasize that this is the same assumption made in GAIL and AIRL. In our experiments, we used 1e4 expert states to train GAIL (we also updated Table 3 and Appendix D.5).
> > >
> > > Moreover, we want to reiterate that our main contribution is not a new algorithm: the idea of matching state marginals was discussed in prior work (e.g., [1]), and (as we show in Section 4) was done implicitly in exploration methods based on predictive error. Our paper contributes an understanding of precisely what these previous exploration methods are doing, and makes the connection between exploration and distribution matching explicit. In some sense, our experiments actually were a false flag, suggesting that we were proposing a new method. The main aim of our experiments was to show that, since all these prior works are approximately optimizing the same objective, all should perform comparably. Our experiments show that State Marginal Matching is a reasonable exploration objective on both simulated and real-world control tasks.
> > >
> > > 3) *State-action distributions*: Yes, there are tasks where exploration in action space is important (e.g., tasks with very large action spaces). Yes, State Marginal Matching is trivial to extend to state-action distributions: we simply modify the policy update (Eq 5) to use $\log p^*(s, a)$ instead of $\log p^*(s)$.
> > >
> > > [1] "Provably Efficient Maximum Entropy Exploration", Hazan et. al. ICML 2019.
> > > http://proceedings.mlr.press/v97/hazan19a/hazan19a.pdf

---

### Official Review · AnonReviewer3 · 2019-10-23
**Official Blind Review #3**

**Rating:** 3

**Review:**

### Summary

This paper proposes to optimize the state marginal distribution to match a target distribution for the purposes of exploration. This target distribution could be uniform or could encode prior knowledge about downstream tasks. This matching can be done by iteratively fitting a density model on the historical data from the replay buffer, and training a policy to maximize the log density ratio between the target distribution and the learned density model. Experiments are performed on two domains: a simulated manipulation task and a real robotic control task. Overall, the paper is well-written.


### Review

Recommendation: weak reject for the reasons below. The main reason is that this paper ignores very similar prior work which is not properly credited.


The algorithm proposed here is very similar to the algorithm proposed in [1]:
- the objective proposed in  equations (1) and (2) is the same as the second objective in Section 3.1 of [1].
- Algorithm 1 here is almost identical to Algorithm 1 in [1]

The work of [1] is only briefly mentioned in the related work section, and from the description there seems to be a fundamental misunderstanding of it.
It says "their proposed algorithm requires an oracle planner and an oracle density model, assumptions that our method will not require".

Making oracle assumptions is a tool for proving theoretical results, not a feature of an algorithm. An oracle can be any subroutine that one has reason to believe works reasonably well, and how well it works or not is captured in its accuracy parameter (usually \epsilon).
They are used to break down a  more complex algorithm into simpler subroutines (called oracles), and deriving a guarantee on the complex algorithm in terms of the quality of the oracles.
For example, [1] assumes a density estimation oracle, which could be instantiated as a kernel density estimator, a VAE, count-based density estimation in the tabular case, etc.
It also assumes a planning oracle, which could be instantiated using any method for learning a policy (PPO, SAC, policy iteration, etc), or some search method if the environment is deterministic.
The accuracy of the oracles are reflected in the \epsilon_0 and \epsilon_1 parameters, which then show up the guarantee for theorem 1.

Theorem 1 of [1] also shows that the entropy of the policy mixture (i.e. replay buffer) matches the maximum entropy over the policies in the policy class, which is one of the main theoretical claims of the work here.

Given this, I don't see this paper as making any new algorithmic or theoretical contributions. On the other hand, [1] had a very limited empirical evaluation and it would be valuable to have a more thorough empirical investigation of this type of method in the literature. This paper partially does that in the sense that they apply more modern methods (VAEs rather than counts/kernel density estimators) on more complex tasks (a simulated manipulation task and a real robot), and their experiments seem well-executed with proper comparisons. However, since the primary contribution of this paper seems to be empirical, I don't think the current experiments on two domains are enough.

I think this paper could be fine for publication with a fairly significant rewrite placing it in the context of prior work, and expanding the experimental section.
My suggestions are to add experiments on several other continuous control tasks (Mujoco/Roboschool) as well as hard exploration Atari games (Montezuma, Freeway, Pitfall etc), to see how well the density estimation works in pixel domains (and the effect of historical averaging). I would be willing to raise my score if these changes can be made within the rebuttal period.


[1] "Provably Efficient Maximum Entropy Exploration", Hazan et. al. ICML 2019.
http://proceedings.mlr.press/v97/hazan19a/hazan19a.pdf


**Experience Assessment:**

I have published one or two papers in this area.

**Review Assessment: Checking Correctness Of Derivations And Theory:**

I assessed the sensibility of the derivations and theory.

**Review Assessment: Checking Correctness Of Experiments:**

I assessed the sensibility of the experiments.

**Review Assessment: Thoroughness In Paper Reading:**

I read the paper at least twice and used my best judgement in assessing the paper.

---

> ### Author Response · Authors · 2019-11-15
> **Author Response**
>
> Thank you for your detailed response. We aim to convince you that, while our paper shares some elements with [Hazan et al], our paper makes a substantial contribution on top of this prior work. We have also added exploration baselines on the real-world manipulation task (Fig. 4b), and observe that our method outperforms these baselines.  (See Revisions comment below.)
>
> Similarities: Similar to [Hazan et al], our paper suggests that a KL divergence between some target state distribution and the policy state marginal distribution be used as an objective for exploration. The algorithms we introduce are quite similar. Consider the special case of [Hazan et al] where the objective R(d) is a KL divergence, KL(d || Q). The reward function suggested by [Hazan et al], $\nabla KL(d || Q)$ is $log Q(s) - log d_\pi(s) + 1$, which is the same reward that we use (the additive constant 1 does not affect the behavior of the optimal policy).
>
> Differences: Our main contribution comes in situating State Marginal Matching in relation to prior work on exploration. More precisely, we show in Section 4 that a large class of exploration bonuses (those based on prediction error) are all maximizing the same objective: marginal state entropy. Said another way, the objective suggested by [Hazan et al] (Section 3.1) was already being (approximately) optimized by existing methods! Nonetheless, our experiments show that an algorithm designed to explicitly maximize this objective (as introduced by [Hazan et al] and ourselves) performs slightly better at maximizing this objective.
>
> Our main contribution is not a new algorithm: the idea of matching state marginals was discussed in prior work (e.g., [Hazan et al]), and (as we show) was done implicitly in exploration methods based on predictive error. Our paper contributes an understanding of precisely what these prior exploration methods are doing. In some sense, our experiments actually were a false flag, suggesting that we were proposing a new method. The main aim of our experiments was to show that, since all these prior works are approximately optimizing the same objective, all should perform comparably. Our experiments show that State Marginal Matching is a reasonable exploration objective on both simulated and real-world control tasks. To the best of our knowledge, our paper is the first to successfully apply an entropy-based exploration algorithm to a real-world robot.
>
> Revisions: We have updated the discussion of Hazan in Section 2 (paragraph 2) and have added a noted the similarities in Section 3 (paragraph 4). We believe these revisions address the concern that Hazan was not reviewed thoroughly enough.

---

### Author Response · Authors · 2019-11-15
**Author Response: Revisions Summary**

We thank the reviewers for their constructive feedback. Following some of the reviewers' suggestions, we have added the following experimental results:

1) We added exploration baseline results (ICM, Pseudocounts, Count) on the real-world manipulation task (Fig 4b). We note that SMM visits a wider range of states than other exploration baselines. We summarize the hyperparameter sweeps for each exploration method in Table 3.

2) We added a GAIL comparison for simulated (Fig 3c) and real-world (Fig 4b) manipulation experiments. GAIL makes slightly different assumptions than State Marginal Matching (e.g., GAIL requires expert trajectories), and we explain how we compare the methods on a level footing in Appendix D.2. In particular, we used states sampled from p*(s) to train GAIL. We also tried restricting the GAIL discriminator input to particular state dimensions (e.g., object position), and also tried different state sampling distributions. Out of these, we used the best GAIL model to compare against the exploration baselines in Figure 3c and 4b.

Other changes:

3) We fixed typos regarding the state dimensions of Manipulation & D'Claw tasks (Appendix D.1, Table 1):
- ManipulationEnv has state dimension 25, not 10 as previously stated.
- D'Claw has state dimension 12 and action dimension 9, not 2 for both.

---

### Decision · Program_Chairs · 2019-12-19

**Decision:**

Reject

**Comment:**

The paper provides a nice approach to optimizing marginals to improve exploration for RL agents.  The reviewers agree that its improvements w.r.t. the state of the art do not merit a publication at ICLR.  Furthermore, additional experimentation is needed for the paper to be complete.